# Availability and geographic access to breast cancer pathology services in Ghana

Matthew D. Price [1,2], Meghan E. Mali[1,3], Adjei Ernest[4], Afua O. D. Abrahams[5], Eric Goold[6], Liz Elvira [1], Florence Dedey[7], Anne F. Rositch[8,9], Raymond R. Price[1,3,10], Edward K. Sutherland [1,8,11]*

1 Center for Global Surgery, The University of Utah, Salt Lake City, UT, United States of America, 2 Department of Surgery, The Johns Hopkins University School of Medicine, Baltimore, MD, United States of America, 3 Department of Surgery, The University of Utah School of Medicine, Salt Lake City, UT, United States of America, 4 Department of Pathology, Komfo Anokye Teaching Hospital, Kumasi, Ghana, 5 Department of Pathology, University of Ghana Medical School, Accra, Ghana, 6 Department of Pathology, The University of Utah School of Medicine, Salt Lake City, UT, United States of America, 7 University of Ghana Medical School, Accra, Ghana, 8 Department of Epidemiology, The Johns Hopkins Bloomberg School of Public Health, Baltimore, MD, United States of America, 9 Diagnostics Division, Hologic, Inc, San Diego, CA, United States of America, 10 Intermountain Health, Salt Lake City, UT, United States of America, 11 Ensign Global College, Kpong, Ghana

* sutherlandmd@yahoo.com

**Data Availability Statement:** The de-identified data set of the hospital facilities has the geographic locations of the hospitals (both private and governmental) in the study and their service infrastructure and resources. Ethical clearance

## Abstract

### Introduction

Breast cancer poses a significant health challenge in Sub-Saharan Africa, particularly in Ghana, where late-stage diagnoses and limited healthcare access contribute to elevated mortality rates. This study focuses on the crucial role of pathology and laboratory medical (PALM) services in the timely diagnosis of breast cancer within Ghana.

### Methods

A cross-sectional survey of hospitals was completed from November 2020 to October 2021, with 94.8% of identified in-country hospitals participating. Pathology service-related parameters assessed included whether pathology was available for the diagnosis of breast cancer on-site or via external referral, the number of pathology personnel, additional breast cancer diagnostic capabilities including estrogen and progesterone and/or HER2 testing, and the time from biopsy to patients receiving their results. Geospatial mapping was used to identify areas of limited access.

### Results

Of the 328 participating hospitals, 136 (41%) reported breast cancer pathology services, with only 6 having on-site capabilities. Pathology personnel, comprising 15 consultants and 15 specialists, were concentrated in major referral centers, particularly in Greater Accra and Kumasi. An assessment of referral patterns suggested that 75% of the population reside within an hour of breast cancer pathology services. Among the 136 hospitals with access to breast cancer pathology, only a limited number reported that results included ER/PR (38%) and HER2 testing (33%).

approval was given by the Ghana Health Services (GHS) Ethical Review Committee dependent that potential harm to subjects in the context of social stigma, cultural, economic, and other aspects are protected in avoidance of possible discriminatory uses by any groups and bodies, and as such no hospital names were even used within the current manuscript. The data-sharing and ownership within the GHS approved ethics study protocol requires the data to be kept under the supervision of the team at Ensign Global College/Center for Global Surgery and available upon request for researchers who meet the criteria for access to confidential data. For access to the data, requests may be sent to the institution at globalsurgery@hsc.utah.edu.

**Funding:** The study was supported by the University of Utah Center for Global Surgery through the Gardner & Holt Grant with no specific grant or award number. Additional funding included the National Cancer Institute (NCI) Grant #T32CA126607 (Price, M). Hologic provided support in the form of salary for Rositch, A. but did not have any additional role in the study design, data collection, and analysis, decision to publish, or preparation of the manuscript. The specific role of this author is articulated in the 'author contribution's' section.

**Competing interests:** AFR is currently an employee of Hologic. The majority of this study was completed while AFR was engaged as a full-time faculty member at the Johns Hopkins Bloomberg School of Public Health; only the revision phase occurred during AFR's employment at Hologic. This does not alter our adherence to PLOS ONE policies on sharing data and materials. There are no patents, products in development or marketed products associated with this research to declare.

## Conclusion

Ghana has been able to ensure significant pathology service availability through robust referral pathways with centralized labs. Despite this, difficulties persist with the majority of pathology results not including hormone receptor testing which is important in providing tumor specific treatment.

## Introduction

Breast cancer remains a pressing global health concern, with its impact particularly pronounced in Sub-Saharan Africa, where late-stage diagnoses and constrained access to healthcare services pose formidable challenges to effective treatment and care [1–4]. The intricate interplay of socio-economic, cultural, and healthcare system factors further exacerbates the predicament, contributing to delayed diagnoses and elevated mortality rates associated with breast cancer in the region [5–7]. Ghana, grapples with these challenges, with breast cancer representing the leading type of cancer diagnosed and the second leading cause of cancer-related death [8–12]. Decreasing the interval from symptom onset to pathologic diagnosis and treatment initiation is essential to improved survival [13, 14]. For this to occur, patients require access to high-quality and timely pathology and laboratory medical (PALM) services [15–17]. The most common barriers hindering the quality of breast cancer pathology cited in the literature include shortcomings in equipment, organization, and insufficiently qualified personnel [15, 18].

PALM services, and pathologists in particular, play a critical role in disease classification and staging through the collection, handling, and examination of breast specimens [15, 19–22]. Immunohistochemistry and molecular testing performed by pathologists also aid in formulating the correct diagnosis and assists in selecting the appropriate hormonal or chemotherapy regimen. Pathologists are then able to collaborate with clinicians to guide treatment decisions, helping to determine prognosis, and ensure comprehensive patient care.

In Ghana, the path to becoming a pathologist requires rigorous academic and practical training. This process includes six years of medical school followed by a mandatory two years of housemanship during which newly graduated doctors serve as general medical practitioners. Subsequently, they become eligible to apply for postgraduate pathology training. Upon securing a postgraduate position, trainees dedicate the next three years to becoming anatomic pathology specialists. At this juncture, they decide to either conclude their training or opt for an additional two-year period to attain fellowship status, equipping them with a broader skill set. Notably, some pathologists acquire training outside of Ghana; however, upon returning, they must successfully pass a licensure examination administered by the Ghana Medical and Dental Council. The amount of rigorous training to become a pathologist along with the general lack of understanding of the role of a pathologist beyond working with the deceased has historically resulted in only a few doctors specializing in pathology. This trend is changing though, and currently there is an increase in the number of doctors choosing pathology as a field of specialization.

Comprehensive data on the number of in-country pathologists in Ghana and accessibility of breast cancer PALM services are limited. In 2008, there was one pathologist for every 2.7 million people in Ghana, and this ratio improved slightly to one pathologist for every 2.2 million people by 2012 [22, 23]. While there is no established standard for the optimal number of pathologists to adequately serve a population, a 2018 Lancet report estimated four pathologists and two clinical scientists would be needed to support a population of 3 to 6 million people

[16, 24]. As an additional comparison, the United States has 65 pathologists per every one million people [25]. A 2014 survey of countries in Sub-Saharan Africa found that Ghana had only two laboratories, both private, that met the standards established by either the 1988 Clinical Laboratory Improvement Amendments (CLIA) or the clinical laboratory standards of the International Organization for Standardization [26–29].

In this study, we aim to identify the number of in-country pathologists throughout Ghana and outline the geographic access to breast cancer PALM services. Through this analysis we seek to identify geographic barriers to PALM services and propose pathways for expansion.

## Methods

### Study design

We retrospectively analyzed a cross-sectional, in-person, hospital-based survey that was administered throughout Ghana from November 2020 to October 2021. Nearly all identified hospitals participated (94.8%). The survey evaluated the availability of breast and cervical cancer services defined by the National Comprehensive Cancer Network (NCCN) Framework for Resource Stratification and included a sub-survey on pathology services [30]. This paper solely focuses on breast cancer pathology services. The full survey design and methods of administration have been previously described in detail by Moustafa, et al. [31] and Schoenhals et al. [32] and was a collaboration between the Ensign Global College in Ghana, the University of Utah Center for Global Surgery, and the Ghana Health Service (GHS). In reporting specific data about the hospitals with on-site pathology, the hospitals were randomly assigned a letter A–F to maintain anonymity.

IRB approval was obtained from the GHS Ethics Review Committee and from each of the teaching hospitals that participated in the study. The need for further IRB approval was waived by the Johns Hopkins School of Public Health as the subject pertains to secondary data analysis involving the use of existing de-identified data. This study pertains to the availability of hospital services and does not pertain to human subject's research. The dataset for this project was accessed between January 1st and March 1st 2023 and again between October 1st and December 31st 2023 for confirmatory analysis.

### Pathology personnel

The number of pathology personnel at each hospital was accounted for in the survey, including the number of pathology consultants and pathology specialists. In Ghana, a pathology consultant is a medically trained pathologist who is generally more senior, has undergone additional training, and has achieved a high level of expertise in the field. A pathology specialist is also a medical professional who has specialized in pathology but is generally more junior with less experience and training than pathology consultants.

### Pathology services

Hospitals were assessed if their patients had access to pathology services for the diagnosis of breast cancer either through an on-site pathology lab or through sending their surgical specimens to external labs (Fig 1). If hospitals indicated they provided pathology services via sending their specimens to an external facility, they were further queried as to which facility or facilities they send their specimens. Additional breast cancer pathology services inquired about included estrogen receptor (ER) and progesterone receptor (PR) immunohistochemistry (IHC) staining, HER2/Neu IHC staining, and HER2/neu fluorescence in situ hybridization (FISH). The average time for patients to receive pathology results from the time of their biopsy was estimated by the survey respondents at each hospital facility. Respondents also estimated if

1. Are pathology services available (either in house or as referral) for diagnosis of breast cancer? (Y1, Y0, N)*
2. If pathology is available, is it performed in the facility, or sent to an outside/referral facility?
   a.  in house
   b.  external/send out/referral
3. If pathology is sent out to an external facility list facility name and location (including country if appropriate)
4. How long does it take for the patient to receive their results?
   a.  < 2 weeks
   b.  > 2 weeks but < 1 month
   c.  > 1 month
5. Is estrogen receptor (ER) immunohistochemistry (IHC) staining performed on the pathology? (Y1, Y0, N)*
6. Is progesterone receptor (PR) IHC staining performed on the pathology? (Y1, Y0, N)*
7. Is HER2/neu IHC staining performed on the pathology? (Y1, Y0, N)*
8. Is HER2/neu fluorescence in situ hybridization (FISH) performed on the pathology? (Y1, Y0, N)*
9. How long does it take for the patient to receive results about the special stains (from questions 5-8)?
   a.  < 2 weeks
   b.  > 2 weeks but < 1 month
   c.  > 1 month

*Y1 = available more than 80% of the time, Y0 = available less than 80% of the time, N = not available

**Fig 1. Sub-survey questions pertaining to breast cancer pathology services.**

the pathology services were available more than 80% of the time in the year preceding the survey, less than 80%, or not available.

## Geospatial mapping

To better visualize and define the geographic access to breast cancer pathology services throughout Ghana, hospitals that reported access to pathology were mapped using ArcGIS Pro Software [33]. The proportion of the population within Euclidean distances of 45 km of hospitals that provide pathologic diagnosis of breast cancer was calculated by incorporating the 2021 World Pop density raster [34]. This distance was chosen as has been done by prior studies as the threshold predictive of being within 1 hour of care [31, 32]. Distance between hospitals that send their specimens to external facilities and the facility where the pathology is analyzed was generated using minimum road distances provided by google maps.

## Results

Out of the 328 hospitals that participated in the survey, 136 (41%) indicated having access to pathology services for breast cancer diagnosis. However, only 6 of these hospitals reported having on-site pathology capabilities, while the remaining 130 relied on sending their specimens

**Table 1.  Distribution of hospital-based breast cancer pathology service availability (on-site and off-site) in Ghana by regions.**

| Region: | Population (% of total) | Hospitals Surveyed (% of total hospitals) | On-site n | External n |
|---|---|---|---|---|
| Ashanti | 6,352,774 (21%) | 70 (21%) | 1 | 37 |
| Greater Accra | 5,318,675 (17%) | 88 (27%) | 2 | 21 |
| Eastern | 3,154,986 (10%) | 34 (11%) | 1 | 10 |
| Central | 2,784,283 (9%) | 21 (6%) | - | 6 |
| Western Region | 2,101,432 (7%) | 18 (6%) | 1 | 17 |
| Northern | 1,999,079 (6%) | 13 (4%) | 1 | - |
| Volta | 1,897,386 (6%) | 16 (5%) | - | 9 |
| Upper East | 1,118,345 (4%) | 11 (3%) | - | - |
| Bono | 1,116,908 (4%) | 11 (3%) | - | 8 |
| Bono East | 1,088,238 (3%) | 12 (4%) | - | 7 |
| Western North | 803,893 (3%) | 10 (3%) | - | 8 |
| Upper West | 774,716 (2%) | 8 (2%) | - | - |
| Oti | 738,791 (2%) | 6 (2%) | - | 3 |
| Ahafo | 627,996 (2%) | 3 (1%) | - | 4 |
| Savannah | 609,894 (2%) | 3 (1%) | - | - |
| North East | 569,475 (2%) | 4 (1%) | - | - |
| TOTAL | 31,732,128 | 328 | 6 | 130 |

to one of the 6 hospitals with on-site pathology or external private pathology labs **Table 1**. These referral patterns have enabled 75% of the country's population to live within 1 hour of a hospital that provides services for breast cancer pathology. Notably, all hospitals equipped with on-site pathology for breast cancer reported the availability of breast cancer diagnosis exceeding 80% of the time throughout the year preceding the survey. In contrast, only 68% (n = 88) of the hospitals that sent their specimens to other facilities reported the availability of breast cancer diagnosis exceeding 80% the year preceding the survey.

## Pathology personnel

There were 30 medically trained pathologists comprising 15 consultants and 15 specialists, distributed across 15 different hospitals. A large proportion of in-country pathologists (45%) were situated in four different hospitals within the Greater Accra Region. Notably, 23% (N = 7) of all pathologists nationwide were employed at a single large public teaching hospital in Accra (Hospital B). The second-largest concentration of pathologists was observed at a large teaching hospital in Kumasi (Hospital A), representing 14% (n = 4) of the total-in-country pathologists **Table 2**.

Nine hospitals reported on-site pathologists but could not provide on-site pathology for breast cancer, opting to send their breast cancer specimens to other facilities or to private pathology labs for analysis (**Fig 2**). Lack of resources for the diagnosis of breast cancer was noted as a common challenge for these hospitals.

## Breast cancer pathology referral pathways

Of the 6 hospitals with on-site breast cancer pathology, three served as primary referral sites for 76% (n = 104) of all surveyed hospitals (**Fig 3**). One of these was a large teaching hospital in Kumasi (Hospital A), which received specimens from 60 other hospitals **Table 2**. This hospital received specimens from as far south as the coast in the Western Region (approx. 290 km), extending to a single hospital in the Northern Region (approx. 391 km). The median distance from external hospital facilities who send their specimens to Hospital A was 22.5 km

**Table 2. Hospitals with on-site breast cancer pathology and summary of availability of breast cancer pathology as reported by hospitals who refer to these centers.**

| Hospital | A | B | C | D | E | F |
|---|---|---|---|---|---|---|
| **Hospital Location** | **Kumasi** | **Accra** | **Takoradi** | **Accra** | **Koforidua** | **Tamale** |
| **Number of pathologists** | 4 | 7 | 2 | 4 | 2 | 2 |
| **Number of external hospitals sending breast cancer specimens to this facility[a]** | 60 | 31 | 10 | 2 | 1 | 0 |
| **Proportion of referral hospitals reporting breast pathology available >80%** | 82% | 55% | 10% | 100% | 100% | N/A |
| **Special testing availability as reported by hospitals who send specimens to this hospitals** | | | | | | |
| ER/PR | 25% | 58% | 100% | 100% | 100% | N/A |
| HER2 IHC | 23% | 28% | 63% | 100% | 0% | N/A |
| HER2 FISH | 21% | 25% | 72% | 100% | 0% | N/A |
| **Distance in km from external hospital to facility specimen is analyzed, Median (IQR)** | 22.5 (4.8–121) | 121 (30–194) | 14.7 (6.2–54.2) | 25 (N/A) | 70 (N/A) | N/A |
| **Proportion of external hospitals who send to this facility reporting specimen results < 2 weeks** | 15% | 19% | 30% | 0% | 0% | N/A |
| **Proportion of external hospitals who send to this facility reporting specimen results <1mo** | 97% | 90% | 80% | 100% | 100% | N/A |

[a]Hospitals could indicate sending specimens to more than one external hospital or private facility, when this occurred, they were counted for both hospitals in this table.

(Interquartile range [IQR]: 4.8–121 km) and the majority (82%) of them reported breast pathology available >80% of the time in the year preceding the survey.

Another primary referral site was a large teaching hospital in Accra (Hospital B), which received specimens from 31 other hospitals, mostly clustered in the southwestern quarter of the country. The median distance from external hospital facilities to Hospital B was greater, 121 km (IQR: 30–194 km), than those that send to Hospital A (p < 0.001) with 55% reporting pathology available >80% of the time the year preceding the survey.

The third major pathology specimen referral hospital was a regional hospital in Takoradi (Hospital C), which received specimens from 10 other hospitals in the region, with a median distance of 14.7 km (IQR 6.2–54.2 km).

Private, non-hospital pathology labs were not included in this study's survey analysis; however, 36 hospitals reported sending breast cancer specimens to these facilities **Table 3**. The most cited private pathology labs were Ghana Standard Authority, Pathologists without Borders, and A.C.T. (Accuracy, Completeness, Timeliness) Diagnostic. Only one hospital reported sending their breast cancer pathology out of the country for diagnosis (to a lab in South Africa). In the Central Region, multiple hospitals reported sending their specimens to the teaching hospital in Cape Coast. However, this hospital reported no access to on-site breast cancer pathology and sent all specimens to a local private pathology lab.

## Additional breast pathology services

While 41% of hospitals reported access to breast cancer pathology services, only 38% of them reported the inclusion of ER/PR testing, and 33% included HER2/Neu testing in their pathology results. Among the three major pathology referral hospitals, those located in Kumasi (Hospital A) and Accra (Hospital B) individually affirmed both ER/PR testing and HER2/neu testing available more than 80% of the time in the year preceding the survey. However, the hospital in Takoradi (Hospital C) reported that all these services were available less than 80% of the time. Despite these services being reportedly accessible at the primary referral centers, a significant proportion of hospitals indicated that their specimens did not include these specific test results.

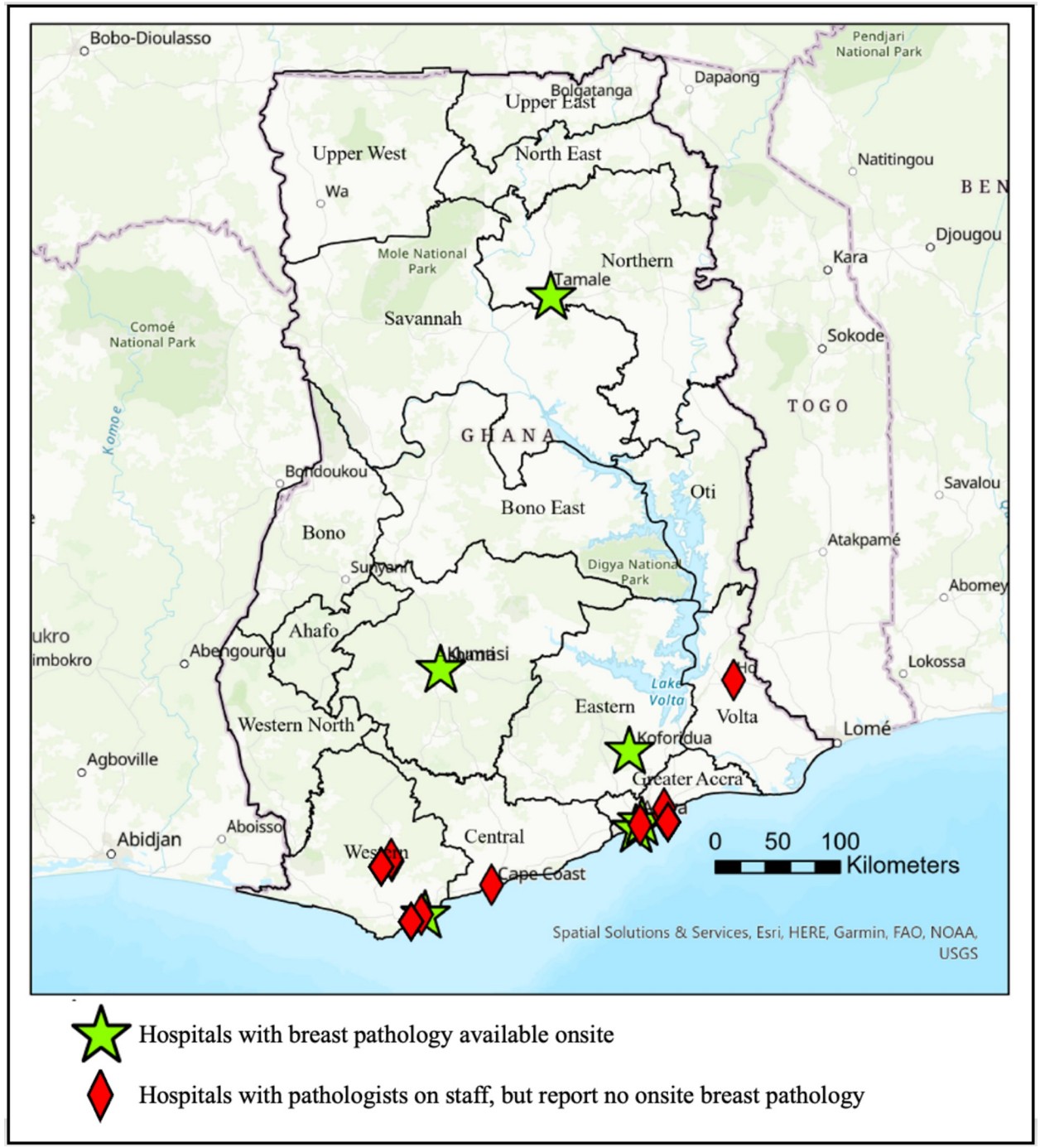

**Fig 2. Geographic location of pathologists and locations of breast cancer pathology in Ghana.** Content is the intellectual property of Esri and is used herein with permission. Copyright © 2024 Esri and its licensors. All rights reserved.

## Time required for results

The majority of hospitals that reported access to breast cancer pathology, whether on-site or via external send-out, obtained pathology results within 1 month of sending the specimen (94%). Results within 2 weeks were less frequent (reported by 26% of hospitals), with results in

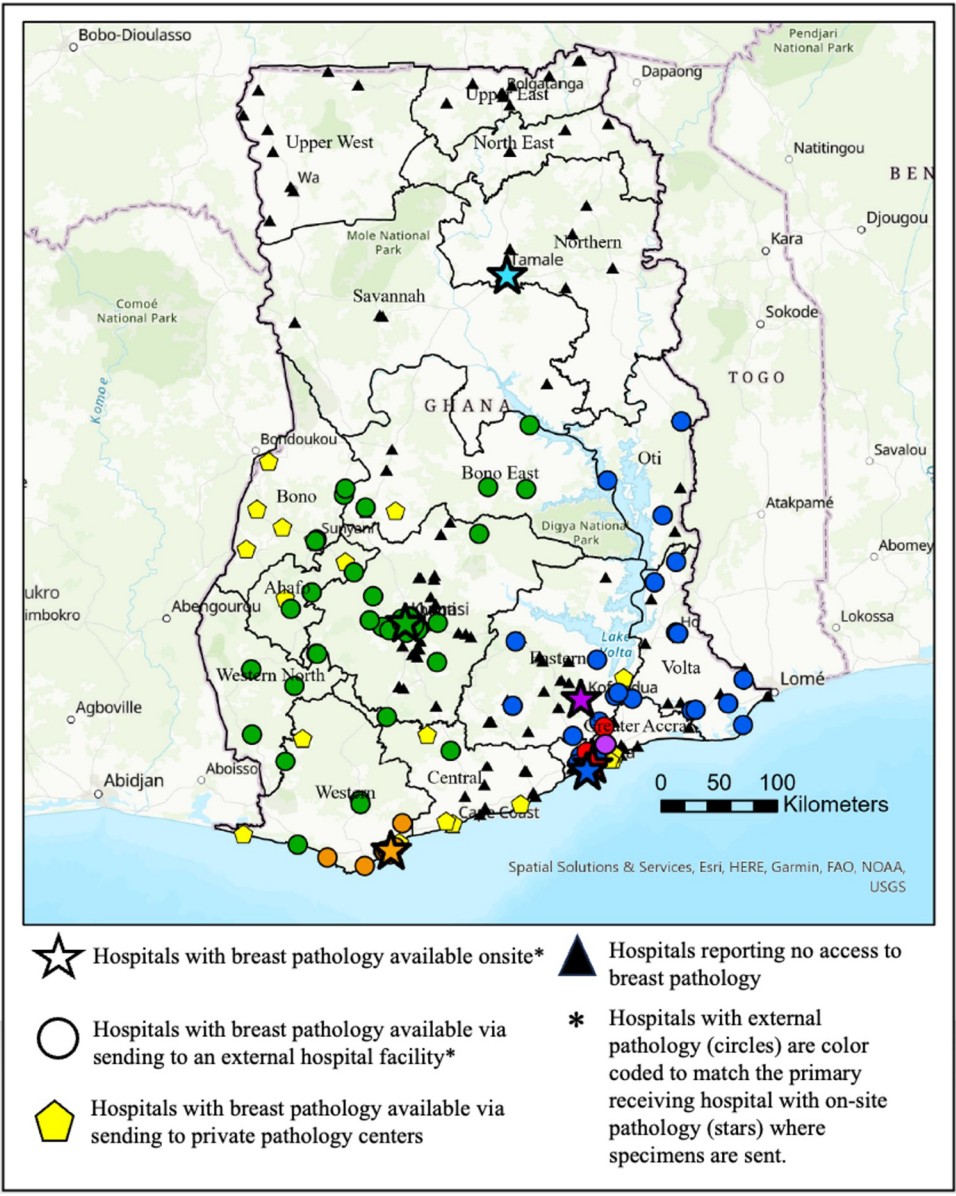

**Fig 3. Breast cancer pathology referral patterns in Ghana.** Content is the intellectual property of Esri and is used herein with permission. Copyright © 2024 Esri and its licensors. All rights reserved.

under 2 weeks more commonly reported by hospitals that sent their specimens to private pathology labs (68% of hospitals) (p < 0.001) **Table 2**.

## Discussion

Access to pathology and laboratory medicine (PALM) is a critical yet challenging aspect in achieving an early diagnosis of breast cancer, particularly in low-resource settings. Inadequate workforce, education, infrastructure, and quality standards hinder optimal PALM service availability [15–17]. National strategic laboratory plans, adequate financing, and leadership roles for pathologists are necessary to overcome these challenges. This study provides a

**Table 3. Summary of availability of breast pathology as reported by hospitals who refer to private pathology labs.**

| | |
|---|---|
| **Number of private pathology labs as indicated by hospitals sending their specimens**[a] | 9 |
| **Number of Hospitals sending breast cancer pathology specimens to private facilities** | 36 |
| **Proportion of referral hospitals reporting breast pathology available >80%** | 66% |
| **Proportion of hospital who report special testing is available at private pathology facilities** | |
| **ER/PR** | 48% |
| **HER2 IHC** | 30% |
| **HER2 FISH** | 41% |
| **Proportion reporting specimen results < 2 weeks** | 68% |
| **Proportion reporting specimen results <1mo** | 92% |

[a]Private pathology labs were not surveyed, this data is derived as reported by hospitals who send to private facilities.

comprehensive assessment of the geographic landscape and availability of breast cancer PALM services and pathology workforce in Ghana.

Our study reveals a notable success in the establishment and utilization of hospital pathology referral pathways for breast cancer diagnosis in Ghana, with a resultant 3 out of 4 people residing within one hour of services. The concentration of pathologists in major centers, particularly in Kumasi and Accra, has facilitated efficient diagnostic referral services. If pathologists were more equitably distributed throughout the country it is unknown if this would result in improved service availability or if this would instead diminish the capacity of the already established referral patterns. A few key hospitals serve as primary referral sites for a substantial portion of the country, creating a strategic distribution that enables a significant proportion of the population to access breast cancer pathology within a reasonable distance. These centralized hubs can help streamline the diagnostic process and contribute to more widespread availability thereby contributing to the potential for early diagnosis and treatment initiation. Sayed et al. and Flemming et al. refer to these large pathology referral centers as Tier 4 laboratories which are typically based out of national or teaching hospitals that receive specimens from their own patients and referrals from lower-tiered facilities [16, 35].

Despite the overall success of the hospital pathology referral pathways, a significant challenge emerges in the form of limited diagnostic information. A considerable number of hospitals, despite having access to pathology services, reported that their breast cancer specimens did not include essential results such as ER/PR and HER2/Neu testing. The absence of these critical pieces of information hinders the ability to tailor treatment strategies based on the specific characteristics of the tumors [18, 24, 36]. Treating breast cancer without knowledge of hormone or HER2 status may lead to suboptimal treatment decisions and compromise patient outcomes [37]. Thus, comprehensive pathology services that include a range of diagnostic tests are necessary for clinicians to provide personalized and effective treatment for breast cancer. If endocrine and/or HER2 targeted therapies are unavailable these studies have no utility, however these targeted therapies are available in Ghana making it essential for the testing to be done [24, 32]. One weakness of the current study is our inability to determine the reasons for the low rates of additional tests (such as ER, PR and HER2) despite the availability of these tests at the larger referral centers. Follow-up studies are therefore needed to ascertain the reasons behind the non-inclusion of these essential laboratory tests at these hospital facilities.

Additionally, the time required for obtaining pathology results is a crucial aspect of breast cancer care and influences patient outcomes. In our study we found that most hospitals reported receiving pathology results within one month, but only 26% reported results within two weeks. While the geographic landscape and workforce distribution are important in

evaluating access to care, rapid turnaround time is essential for timely initiation of treatment [13, 14]. It should be noted however that the times to results in our study are not patient reported, but rather estimates provided by the hospital who responded to the survey. Further in-depth analysis is needed to understand the specific challenges leading to delays.

While the centralized referral system has successfully provided access for most of the population, there are still areas, particularly in the Northern Regions, where access remains limited. Further referral pathways should be explored to expand access to these hospitals, particularly given that one of the six hospitals in the country with on-site pathology is a teaching hospital in the Northern Region. Service expansion, however, can be challenging without expanded resources and an increased workforce. Additionally, there are nine hospitals with pathologists on staff that do not have the capability to provide breast cancer pathology. These hospitals could serve as potential targets for breast cancer pathology service expansion.

Estimates provided by the chief pathologist at Korle Bu Teaching Hospital 20 years ago noted that the two major teaching hospitals should have 8 to 12 pathologists and each district and regional hospital should have 1–2 pathologists to be able to provide appropriate pathology coverage [38]. Prior studies have also provided estimates on the number of pathologists in Ghana with nine pathologists estimated to be in-country in the year 2008 and up to 12 in 2012 [22, 23]. Through this study, we show that there are at least 30 medically trained pathologists distributed across 15 hospital facilities which equates to 1 pathologist per every 1,056,961 million people. While this shows significant growth from prior estimates, it also highlights the need for further investment in expanding the pathology workforce in Ghana to provide optimal population-level care.

Despite the valuable insights provided by this study, certain limitations must be acknowledged. Firstly, the data used for this analysis is based on a survey conducted just over 2 years ago. During this time, changes in service availability and the number of pathologists may have occurred, potentially impacting the current landscape of breast cancer pathology services. While the in-person surveys were with the most knowledgeable individual on breast cancer services at each hospital and additional providers were consulted where information was missing, there could remain errors related to respondents' knowledge or lack thereof. Additionally, the study did not include a survey of private pathology laboratories, limiting the identification of all pathologists in the country, though multiple pathologists have dual appointments at local hospitals and with private pathology labs. Future research should consider more frequent updates and comprehensive inclusion of all relevant facilities to provide a more accurate and up-to-date picture of breast cancer pathology services. Additionally, given the survey approach, these results do not reflect the true patient-level coverage of services or time to results. Rather, our results reflect general availability of services. However, this analysis remains the most comprehensive assessment on the distribution of pathology services in Ghana to date with 95% of hospitals having participated in the survey.

In conclusion, this study offers a comprehensive examination of breast cancer pathology services in Ghana, shedding light on both achievements and obstacles. While the centralized referral system has successfully increased accessibility, challenges persist, including limited diagnostic information and geographical disparities. To address these issues, ongoing efforts are needed to enhance pathology services by incorporating essential diagnostic tests and expanding referral pathways in the northern part of the country. Future research is needed to understand the true patient level coverage of pathology services, rather than solely what is reported at the hospital-level. Additionally future research should prioritize real-time monitoring of service availability, sustained collaboration with private pathology labs, and exploration of innovative approaches to enhance breast cancer diagnosis and treatment outcomes in Ghana. Ultimately, a multidimensional and collaborative approach is crucial to overcoming

the complexities associated with breast cancer in the region and improving the overall landscape of pathology services.

## Acknowledgments

We would like to acknowledge the following persons and institutions for their immense contributions towards the success of the project: Dr. Alberta Biritwum-Nyarko and Mrs Irina Ofei both of the Ghana Health Services; the Director General of the Ghana Health Services, Dr. Patrick Kuma Aboagye; the Health Facilities Regulatory Agency, Ghana; the Ensign Global College; Dr. Moustafa Moustafa; Dr. Ousman Sanyang, Dr. Grace Ayensu-Danquah and Mr. Jonathan Nellermoe.

## Author Contributions

**Conceptualization:** Matthew D. Price, Meghan E. Mali, Adjei Ernest, Afua O. D. Abrahams, Eric Goold, Liz Elvira, Florence Dedey, Anne F. Rositch, Raymond R. Price, Edward K. Sutherland.

**Data curation:** Meghan E. Mali, Florence Dedey, Raymond R. Price, Edward K. Sutherland.

**Formal analysis:** Matthew D. Price, Meghan E. Mali, Raymond R. Price, Edward K. Sutherland.

**Investigation:** Matthew D. Price, Meghan E. Mali, Anne F. Rositch, Raymond R. Price, Edward K. Sutherland.

**Methodology:** Matthew D. Price, Meghan E. Mali, Adjei Ernest, Eric Goold, Liz Elvira, Florence Dedey, Anne F. Rositch, Raymond R. Price, Edward K. Sutherland.

**Project administration:** Matthew D. Price, Raymond R. Price, Edward K. Sutherland.

**Resources:** Matthew D. Price, Adjei Ernest, Edward K. Sutherland.

**Software:** Matthew D. Price, Edward K. Sutherland.

**Supervision:** Matthew D. Price, Adjei Ernest, Afua O. D. Abrahams, Raymond R. Price, Edward K. Sutherland.

**Validation:** Matthew D. Price, Afua O. D. Abrahams, Eric Goold, Florence Dedey, Edward K. Sutherland.

**Visualization:** Matthew D. Price, Adjei Ernest, Eric Goold, Anne F. Rositch, Raymond R. Price, Edward K. Sutherland.

**Writing – original draft:** Matthew D. Price.

**Writing – review & editing:** Matthew D. Price, Meghan E. Mali, Adjei Ernest, Afua O. D. Abrahams, Eric Goold, Liz Elvira, Florence Dedey, Anne F. Rositch, Raymond R. Price, Edward K. Sutherland.

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
