## [Decision Letter · Decision Letter 0]

6 May 2024

PONE-D-24-04284Availability and Geographic Access to Breast Cancer Pathology Services in GhanaPLOS ONE

Dear Dr. Sutherland,

Thank you for submitting your manuscript to PLOS ONE. After careful consideration, we feel that it has merit but does not fully meet PLOS ONE’s publication criteria as it currently stands. Therefore, we invite you to submit a revised version of the manuscript that addresses the points raised during the review process.

**Congratulations to the authors for a well written manuscript. Further comments can be found under "Additional Editor Comments". **

We look forward to receiving your revised manuscript.

Kind regards,

Kekeli Kodjo Adanu, MB CHB, MPH

Academic Editor

PLOS ONE

Journal Requirements:

The study was supported by the University of Utah Center for Global Surgery through the Gardner & Holt Grant with no specific grant or award number. Additional funding included the National Cancer Institute (NCI) Grant # T32CA126607 (Price, M). We would also like to acknowledge the following persons and institutions for their immense contributions towards the success of the project: Dr. Alberta Biritwum-Nyarko and Mrs Irina Ofei both of the Ghana Health Services; the Director General of the Ghana Health Services, Dr. Patrick Kuma Aboagye; the Health Facilities Regulatory Agency, Ghana; the Ensign Global College; Dr. Moustafa Moustafa; Dr. Ousman Sanyang, Dr. Grace Ayensu-Danquah and Mr. Jonathan Nellermoe.

Although AFR is employed by Hologic, she maintained her role on this project under her Adjunct appointment at Johns Hopkins Bloomberg School of Public Health.  The remaining authors have no relevant conflicts of interest to disclose.

We note that one or more of the authors are employed by a commercial company: Hologic and Johns Hopkins Bloomberg School of Public Health 

“The funder provided support in the form of salaries for authors, but did not have any additional role in the study design, data collection and analysis, decision to publish, or preparation of the manuscript. The specific roles of these authors are articulated in the ‘author contributions’ section.”

5. We note that Figures 2 and 3 in your submission contain map images which may be copyrighted. All PLOS content is published under the Creative Commons Attribution License (CC BY 4.0), which means that the manuscript, images, and Supporting Information files will be freely available online, and any third party is permitted to access, download, copy, distribute, and use these materials in any way, even commercially, with proper attribution. For these reasons, we cannot publish previously copyrighted maps or satellite images created using proprietary data, such as Google software (Google Maps, Street View, and Earth). For more information, see our copyright guidelines: http://journals.plos.org/plosone/s/licenses-and-copyright.

We require you to either present written permission from the copyright holder to publish these figures specifically under the CC BY 4.0 license, or remove the figures from your submission:

a. You may seek permission from the original copyright holder of Figures 2 and 3 to publish the content specifically under the CC BY 4.0 license.  

Additional Editor Comments:

Congratulations to the Authors for a well written paper.

However, there are some few issues that require clarification.

Introduction - The authors laid a good background for the study and the rationale for the study was adequately explained.

Discussion - The authors made the claim that the concentration of pathologist in major referral centers facilitated efficient diagnostic services. Please justify this. Will a more equitable distribution of pathologist in-country rather lead to a more efficient diagnostic delivery system? Please clarify this further in your discussion.

With regards to the low reportage of ER/PR testing and Her-2/neu testing by health facilities, in spite of the availability of these tests at the referral centers, please clarify the following

1. Who were the respondents at the various health facilities?

2. Are these respondents health professionals with adequate knowledge/information on these tests and what they meant?

3. Will the knowledge or lack thereof of respondents influence the outcome obtained and could that be a possible limitation of your study?

Reviewers' comments:

Reviewer's Responses to Questions

**Comments to the Author**

1. Is the manuscript technically sound, and do the data support the conclusions?

Reviewer #1: Yes

Reviewer #2: Yes

2. Has the statistical analysis been performed appropriately and rigorously? 

Reviewer #1: Yes

Reviewer #2: Yes

3. Have the authors made all data underlying the findings in their manuscript fully available?

Reviewer #1: Yes

Reviewer #2: Yes

4. Is the manuscript presented in an intelligible fashion and written in standard English?

Reviewer #1: Yes

Reviewer #2: Yes

5. Review Comments to the Author

**Reviewer #1:** This is a cross-sectional survey of hospitals that assess pathology service- related parameters, including whether breast pathology was available on-site or via external referral, the number of pathology consultants and specialists, the availability of other diagnostic modealities including ER, PR and HER2 testing, and the time taken for the biopsy results to be ready.

The number of hospitals included (328 hospitals) and the duration (12 months) are ok for the data to be statistically of value.

The Geospatial mapping used to identify areas of limited access is informative and attractive.

The survey questions are adequate and clear.

The methodology in general is written orderly and with clear language.

**Reviewer #2:** The manuscript is well-written, and the data are clearly presented.

The conclusions of the study are clearly stated, and the discussion is comprehensive.

The authors may need to revise the manuscript for the very infrequent typo errors.

6. PLOS authors have the option to publish the peer review history of their article (what does this mean?). If published, this will include your full peer review and any attached files.

Reviewer #1: **Yes: **Dalia Abd El-Kareem

Reviewer #2: **Yes: **Prof. Dr. Amal Abd El hafez

---

## [Author Response · Author response to Decision Letter 0]

1 Jun 2024

Thank you very much for your review of our original research article entitled “Availability and Geographic Access to Breast Cancer Pathology Services in Ghana”. We again confirm that this work is original and has not been published elsewhere, nor is it currently under consideration for publication elsewhere. Please find the responses to the specific items highlighted from the review below:

1. When submitting your revision, we need you to ensure that your manuscript meets PLOS ONE’s style requirements.

a. The style templates have been reviewed and our manuscript has been adjusted accordingly.

a. Thank you for pointing this out. The correct grant information has been included in the ‘Funding information’ section:

i. Gardner & Holt Grant with no specific grant or award number

ii. National Cancer Institute (NCI) Grant #T32CA126607 (Price, M).

3. Please remove funding information from the acknowledgements section or other areas of your manuscript. We will only publish funding information present in the funding statement section of the online submission form. Please remove any funding-related text from the manuscript and let us know how you would like to update your funding statement. Currently, your funding statement reads as follows: The author(s) received no specific funding for this work. Please include your amended statements within your cover letter; we will change the online submission form on your behalf. 

a. Funding information has been removed from the acknowledgements section

b. Our amended funding statement we would like to have updated is as follows:

i. The study was supported by the University of Utah Center for Global Surgery through the Gardner & Holt Grant with no specific grant or award number. Additional funding included the National Cancer Institute (NCI) Grant #T32CA126607 (Price, M). Hologic provided support in the form of salary for Rositch, A. but did not have any additional role in the study design, data collection, and analysis, decision to publish, or preparation of the manuscript. The specific role of this author is articulated in the ‘author contribution’s’ section.

4. Thank you for stating AFR’s employment at Hologic in the competing interest’s section.

a. Please provide an amended funding statement declaring this commercial affiliation as well as a statement regarding the Role of Funders in your study

i. This has been added to the funding statement in part 3 above. 

b. Please also provide an updated competing interests statement declaring this commercial affiliation along with any other relevant declarations relating to employment, consultancy, patents, products in development, or marketed products, etc. Within your competing interest’s statement, please confirm that this commercial affiliation does not alter your adherence to all PLOS ONE policies on sharing data and materials. Please include both an updated funding statement and competing interests’ statement in your cover letter. We will change the online submission form on your behalf. 

i. A. Rositch, a co-author, is currently an employee of Hologic, however Hologic had no role in the study design, data collection, and analysis, decision to publish, or preparation of the manuscript. The majority of this study was completed while A. Rositch was engaged as a full-time faculty member at the Johns Hopkins Bloomberg School of Public Health, only the manuscript co-author revision phase occurred while A. Rositch was an employee at Hologic. This does not alter our adherence to PLOS ONE policies on sharing data and materials.

5. We note that Figures 2 and 3 in your submission contain map images which may be copyrighted. All PLOS content is published under the CC BY 4.0, which means that the manuscript, images and supporting information files will be freely available online, and any third party is permitted to access, download, copy, distribute, and use these materials in any way, even commercially, with proper attribution. For these reasons, we cannot publish previously copyrighted maps or satellite images created using proprietary data, such as google software (Google maps, street view, and earth). We require you to either present written permission from the copyright holder to publish these figures specifically under the CC BY 4.0 license, or remove the figures from your submission;

a. Figures 2 and 3 were generated in Arc GIS Pro by our research team using the Esri Arc GIS pro base maps “world topographic map”. We have obtained written permission from Esri for publication under the CC BY 4.0 license and this documentation is submitted with this resubmission. 

6. Please review your reference list to ensure that it is complete and correct. If you have cited papers that have been retracted, please include the rationale for doing so in the manuscript text or remove these references and replace them with relevant current references. Any changes to the reference list should be mentioned in the rebuttal letter that accompanies your revised manuscript. If you need to cite a retracted article, indicate the article’s retracted status in the References list and include a citation and full reference for the retraction notice. 

a. Our reference list is accurate and up to date with no references cited that have been retracted.

b. The following two references have been added for completeness.

i. 33. Esri. "Topographic" [basemap]. Scale Not Given. "World Topographic Map". December, 2023. http://www.arcgis.com/home/item.html?id=30e5fe3149c34df1ba922e6f5bbf808f.

ii. 34. WorldPop. WorldPop 2021 Population Density Raster [Internet]. WorldPop; 2021 [December 2023]]. Available from: [www.worldpop.org].

7. Response to editor’s and reviewers’ comments

a. Discussion – the authors made the claim that the concentration of pathologist in major referral centers facilitated efficient diagnostic services. Please justify this. Will a more equitable distribution of pathologist in-country rather lead to a more efficient diagnostic delivery system? Please clarify this further in your discussion.

i. We agree with your reasoning. Our intention with this sentence/paragraph was to highlight the success of such widespread referral pathways, and such widespread capacity for referrals is only possible with a larger concentration of pathologists. We have re-arranged the sentence and added an additional sentence to now read as follows:

1. The concentration of pathologists in major centers, particularly in Kumasi and Accra, has facilitated efficient diagnostic referral services. If pathologists were more equitably distributed throughout the country it is unknown if this would result in improved service availability or if this would instead diminish the capacity of the already established referral patterns.

b. With Regards to the low reportage of ER/PR testing and HER2/Neu testing by health facilities, in spite of the availability of these tests at the referral centers, please clarify the following:

i. Who were the respondents at the various health facilities?

1. The more in-depth methodology of how and to whom the survey was administered is outlined in the other papers that are cited in our methods section. As to the respondents for the survey, they were “key administrative personnel, the most knowledgeable clinical specialist (eg, medical director, hospital superintendent) of each facility, or the lead breast cancer specialist. If a question was encountered that the respondent did not know, the appropriate person within the hospital was contacted.” The following sentence in the methods section has been updated to include the words “and methods of administration”.

a. The full survey design and methods of administration have been previously described in detail by Moustafa, et. al. [31] and Schoenhals et. al. [32]

ii. Are these respondents health professionals with adequate knowledge/information on these tests and what they meant?

1. See response above. In many cases yes, and when not, or the information was not known by the respondent health professionals with the knowledge were contacted.

iii. Will the knowledge or lack thereof of respondents influence the outcome obtained and could that be a possible limitation of your study?

1. Absolutely, with any human response we would expect there be to some level of error, for this we have added the following sentence in the limitations section of our manuscript. 

a. While the in-person surveys were with the most knowledgeable individual on breast cancer services at each hospital and additional providers were consulted where information was missing, there could remain errors related to respondents’ knowledge or lack thereof.

---

## [Editor Report · Decision Letter 1]

7 Jun 2024

Availability and Geographic Access to Breast Cancer Pathology Services in Ghana

PONE-D-24-04284R1

Dear Dr. Sutherland,

We’re pleased to inform you that your manuscript has been judged scientifically suitable for publication and will be formally accepted for publication once it meets all outstanding technical requirements.

Kind regards,

Kekeli Kodjo Adanu, MB CHB, MPH

Academic Editor

PLOS ONE

Additional Editor Comments (optional):

Congratulations on this detailed and well written research on the availability of breast cancer pathology services in Ghana. This adds considerably to the scarce body of knowledge on the subject matter. Congratulations once again.
---

## [Editor Report · Acceptance letter]

5 Jul 2024

PONE-D-24-04284R1 

PLOS ONE

Dear Dr. Sutherland, 

I'm pleased to inform you that your manuscript has been deemed suitable for publication in PLOS ONE. Congratulations! Your manuscript is now being handed over to our production team.

Kind regards, 

on behalf of

Dr. Kekeli Kodjo Adanu 

Academic Editor

PLOS ONE